# Regulation of Fibrotic Processes in the Liver by ADAM Proteases

**DOI:** 10.3390/cells8101226

**Published:** 2019-10-09

**Authors:** Dirk Schmidt-Arras, Stefan Rose-John

**Affiliations:** Christian-Albrechts-University Kiel, Institute of Biochemistry, 24118 Kiel, Germany; rosejohn@biochem.uni-kiel.de

**Keywords:** ADAM, protease, EGFR, hepatic stellate cell, liver fibrosis

## Abstract

Fibrosis in the liver is mainly associated with the activation of hepatic stellate cells (HSCs). Both activation and clearance of HSCs can be mediated by ligand–receptor interactions. Members of the a disintegrin and metalloprotease (ADAM) family are involved in the proteolytic release of membrane-bound ligands and receptor ectodomains and the remodelling of the extracellular matrix. ADAM proteases are therefore major regulators of intercellular signalling pathways. In the present review we discuss how ADAM proteases modulate pro- and anti-fibrotic processes and how ADAM proteases might be harnessed therapeutically in the future.

## 1. Introduction

The liver consists of different cell types including hepatocytes, cholangiocytes, Kupffer cells, liver sinusoidal endothelial cells (LSECs), and hepatic stellate cells (HSCs). The vast majority of liver resident cells is made up by hepatocytes, also termed liver parenchymal cells. Approximately one tenth of the total number of resident liver cells are HSCs that reside in the space of Disse, a location that is lined by LSECs on one side and hepatocytes on the other side. Under physiological conditions, HSCs are described to maintain a quiescent and non-proliferative state and are mainly characterized by the storage of lipid droplets containing retinyl esters. HSCs are discussed to be professional antigen presenting cells, in particular presenting lipid antigens to natural killer T-(NKT) cells [1,2].

Liver fibrosis is the net accumulation of extracellular matrix (ECM) or scar tissue in the liver. HSCs have been described to be the major source of ECM in liver fibrosis. As a consequence of liver damage, HSCs become activated and trans-differentiate into myofibroblasts that have a proliferative, contractile, and inflammatory phenotype. Activated HSCs are characterised by (among other aspects) the presence of the cytoskeletal proteins alpha smooth muscle actin (αSMA), glial fibrillary acidic protein (GFAP), desmin, platelet-derived growth factor receptor-β (PDGFRβ), the enzyme lecithin retinol acyltransferase (LRAT), and the massive expression of collagen I [3]. Activation of HSCs occurs as a consequence of multiple intra- and intercellular signalling cues [2]. Some of them will be discussed in more detail below. Limited proteolysis reactions by a disintegrin and metalloproteases (ADAMs) are key events in several paracrine signalling pathways [4]. Therefore, ADAM proteases might also represent master-switches during hepatic fibrosis. Here, we discuss known and potential fibrosis-associated pathways regulated by ADAM proteases and review the current knowledge on ADAM protease implication in hepatic fibrosis.

## 2. ADAM Proteases

The superfamily of zinc-containing proteases, termed metzincins, is characterised by the presence of an invariant HEXXHXXGXXH zinc-binding motif within the protease domain [5,6]. Metzincins comprise the four subfamilies matrixin, adamalysins, astacins, and bacterial serralysins. The snake venom metalloproteinases (SVMPs), the a disintegrin and metalloproteinases (ADAMs), and ADAMs containing thrombospondin motifs (ADAMTS) build the adamalysin subfamily [7].

The human genome encodes 22 ADAM proteins, of which 10 are considered to be proteolytically inactive [4]. Enzymatically inactive ADAM proteases are thought to be involved in protein folding and protein–protein interactions.

ADAM proteases share an overall domain structure consisting of an N-terminal inhibitory pro-domain, a catalytic metalloprotease domain, followed by a disintegrin domain with a cysteine-rich region, an epidermal growth factor (EGF)-like domain, and finally a transmembrane domain and a cytoplasmic tail (Figure 1a). The family members ADAM10 and 17 are atypical as they are lacking an EGF-like domain. ADAM proteins are synthesised into the endoplasmic reticulum as catalytically inactive proenzymes. Beside its chaperoning function, the N-terminal pro-domain interferes with the catalytic Zn^2+^-ion and thereby inhibits catalytic activity. Within the Golgi apparatus, proteolytic cleavage, e.g., by the Furin protease, removes the N-terminal pro-domain. For the family members ADAM8 and ADAM28, auto-catalytic removal of the pro-domain was demonstrated [8,9]. Therefore it is common sense that pro-domain removal in ADAM proteases is a prerequisite for full activation of catalytic activity. However, we previously challenged this concept in in vitro experiments using overexpression of Furin-resistent ADAM17 variants. These variants were still able to release tumour necrosis factor α (TNFα) from the cell surface [10]. Interestingly, for ADAM9, 10, and 17 an additional pro-protein convertase cleavage site (upstream site) N-terminal to the previously identified Furin cleavage site (boundary site) was discovered [11]. While cleavage of the C-terminal boundary site by Furin was sufficient for pro-domain removal but dispensable for protease activity, as observed in our experiments, cleavage at the N-terminal upstream site was necessary for the dissociation of the pro-domain and full activation of ADAM proteases [11].

The catalytic domain of ADAM proteases adopts an overall structure which is common to all metzincins members. The catalytic cleft is positioned between an N-terminal (NSD) and a C-terminal subdomain (CSD) [6,12].

The catalytic domain is anchored by a five-stranded β-sheet which lines the upper side of the catalytic cleft. The lower side of the catalytic cleft is lined by a central α-helix which contains the HExxH motif that supplies two of the histidines coordinating the Zn^2+^ ion, as well as the glutamate residue that contributes to catalysis. It is followed by a conserved methionine turn that packs against the zinc-binding site [6,12]. Cleavage specificity is determined by the substrate binding pockets. While substrate binding pockets are denoted by S, the substrate residues are denoted by P. Addition of a prime symbol indicates that the position lies in the C-terminal of the cleavage site. For the family members ADAM10 and ADAM17, binding pockets S3, S1, and S1′ determine protease-specific substrate binding. The largest differences between substrate peptides recognized by ADAM10 or ADAM17 are at the P1′ position. While ADAM10 has a deep hydrophobic S1′ pocket and therefore a strong preference for large hydrophobic residues at P1′, the S1′ pocket of ADAM17 is shallower and constrained by an alanine and a valine residue and therefore ADAM17 prefers smaller, non-aromatic hydrophobic residues at P1′ [12].

Lying adjacent to the metalloprotease domain, the disintegrin domain and the cysteine-rich domain seem to be involved in autoregulation. In the absence of substrate binding, these domains fold back to the catalytic centre and therefore limit the access to the substrate specificity pocket of the catalytic site [12,13]. Protein crystallography human ADAM10 has revealed that residues that form the interaction surface between catalytic domain and the cysteine-rich domain are evolutionarily conserved. Furthermore, while overexpression of ADAM10 devoid of its catalytic domain results in dominant negative inhibition of ADAM10 catalytic activity [12], binding of the recently described antibody 8C7 [14] to the ADAM10 cysteine-rich domain releases auto-inhibition and elevates ADAM10 catalytic activity [12]. These data further reinforce the notion that disintegrin and cysteine-rich domains form an auto-inhibitory module forcing ADAM10 into a closed, inactive conformation (Figure 1b). Disintegrin domain and cysteine-rich domain are also involved in binding to substrate proteins. It is speculated that substrate binding releases the auto-inhibitory conformation of disintegrin and cysteine-rich domain (Figure 1b). Interestingly, all human ADAMs contain a hyper-variable region within the cysteine-rich domain with low sequence homology which may be critical for substrate binding and might contribute to the substrate selectivity of ADAM family members [13]. The fact that binding of the endogenous inhibitor tissue inhibitor of metalloproteases 3 (TIMP3) to the catalytic site of the recombinant isolated catalytic domain of ADAM17 is faster in the absence of disintegrin domain and cysteine-rich domain [15] supports the hypothesis that ADAM17 has a similar auto-inhibitory mechanism as ADAM10.

Protease activity of ADAMs can either be constitutive or activated by extracellular and intracellular signalling pathways. Members of the membrane multi-pass protein families inactive rhomboids (iRhoms) [16,17] and tetraspanins (Tspans) [18,19] have been demonstrated to promote maturation of ADAMs on the secretory pathway.

The rhomboids are a family of evolutionarily conserved multi-transmembrane proteins. The non-protease rhomboids lack amino-acid residues that are essential for protease activity. In particular the members iRhom1 and iRhom2 were identified to interact with ADAM17 as early as in the endoplasmic reticulum in order to promote trafficking to the Golgi [16,17]. iRhom2 was demonstrated to be essential for ADAM17 activities such as TNFα release, particularly in myeloid cells [16,17]. iRhoms are furthermore discussed to regulate substrate selectivity of ADAM17 [20].

Tspans are a family of small and evolutionary conserved proteins that span the membrane four times, resulting in the formation of a short and a large extracellular loop. The latter is involved in particular in client protein interaction. Tspans have the ability to form large interacting networks, the so-called tetraspanin-enriched membrane microdomains (TEM) which represent a signalling platform similar to lipid rafts [21]. Previously, it was demonstrated that in particular members of the TspanC8 family, that is characterised by the presence of eight cysteine residues within the large extracellular loop, bind to ADAM10 within the endoplasmic reticulum and promote its trafficking to the plasma membrane [18,19].

The cytoplasmic tail of ADAM proteases is discussed to regulate catalytic activity of some family members (Figure 1c). It contains several sites that can be phosphorylated by diverse cytoplasmic kinases such as protein kinase C (PKC) [22,23], extracellular regulated kinase (ERK) [24,25], and p38 [26]. In particular, engagement of G-protein coupled receptors (GPCRs) that are coupled to an αq/11 subunit results in the downstream activation of phospholipase C (PLC), an increase in the second messengers diacylglycerol (DAG), inositol triphosphate (IP3), and Ca^2+^, and subsequent activation of PKCs [27,28,29]. While activation of ADAMs by PKCs is less well understood, activation of ADAM17 through C-terminal phosphorylation by ERK and p38 was shown to disrupt ADAM17 dimerization and thereby to release the inhibitory binding of tissue inhibitor of metalloproteases (TIMP) 3 [30].

Recently, basic residues within the membrane-proximal cysteine-rich domain of ADAM17 have been claimed to confer binding to phosphatidylserine (PS) [31]. Only under conditions of cell death, such as apoptosis and necroptosis, is PS exposed to the outer leaflet of the plasma membrane. Under apoptotic conditions, PS exposure and binding to ADAM17 cysteine-rich membrane proximal domain (MPD) alter ADAM17 conformation, releasing autoinhibition and inducing cleavage of substrate molecules [31,32]. The MPD of ADAM17 has been discussed to undergo an isomerization of disulfide bridges and therefore to either adopt an “open” or a “closed” conformation [33]. PS binding to isolated recombinant MPD only occurred when MPD was present in an “open” conformation [31], suggesting that ADAM17 activation is regulated at two levels. Interestingly, PS-binding seems to be a prerequisite for ADAM17 activation in general. Therefore, transient PS exposure might not only be restricted to apoptotic conditions.

The C-terminus of ADAM proteases is furthermore important for their negative regulation via endocytosis. Binding of the cytosolic adaptor proteins Grb2 to ADAM12 and AP2 to ADAM10 was demonstrated to mediate internalisation via the clathrin-dependent pathway [34,35]. Interestingly, overexpression of dominant negative dynamin K44A prevented internalisation but enhanced proteolytic processing of surface-bound ADAM10 [36]. ADAM17 was also shown to be internalised upon phorbol-12-myristate-13-acetate (PMA)-stimulation [37]. Complex formation of ADAM17 with iRhoms and the FERM domain-containing protein (FRMD) 8 at the cell surface was identified to prevent ADAM17 internalisation and lysosomal degradation [38,39]. Once internalised, interaction of ADAM17 with the sorting protein phosphofurin acidic cluster sorting protein (PACS) 2 at early endosomes promotes recycling of ADAM17 back to the cell surface [40].

There is some evidence that in the liver, ADAM protease activity is regulated by bile acids. Ursodeoxycholic acid was shown to inhibit PMA-induced ADAM17 activity on both the HepG2 hepatocellular cell line and the hepatic stellate cell line LX-2 [41]. In contrast, deoxycholic acid (DCA) enhanced ADAM17 catalytic activity and release of amphiregulin in colorectal and pancreatic cancer cell lines. This effect was mediated via binding of DCA to the G protein-coupled bile acid receptor 1, also termed TGR5, and subsequent c-Src activation [42].

## 3. Regulation of Pro- and Anti-Fibrotic Signalling Pathways by ADAM Proteases

The proteolytic release of transmembrane protein ectodomains has been termed “shedding”. Shedding is involved in the release of transmembrane signalling molecules, such as cytokines and growth factors, and the removal of receptor ectodomains. The latter one can either be inhibitory or induce paracrine signalling on neighbouring cells. ADAM proteases play a major role in transmembrane protein shedding, which is a prerequisite for regulated intramembrane proteolysis (RIP) [7]. ADAM proteases might therefore regulate liver fibrosis at different levels: (1) release of paracrine-acting cytokines and growth factors, (2) shedding of receptors on HSC membrane, and (3) modulation of ECM.

### 3.1. Regulation of Liver Epithelial Cell Regeneration

Epithelial cell (i.e., hepatocyte or cholangiocyte) damage is common to almost all liver insults. Hepatocyte apoptotic bodies and damage-associated patterns (DAMPs) lead to rapid activation of Kupffer cells (KCs), a liver-resident macrophage population. In these cells, the proteolytic release of membrane-bound pro-inflammatory cytokines such as TNFα and proliferative signals such as cleavage of membrane-bound epidermal growth factor receptor (EGFR) ligands are most likely orchestrated by ADAM17. TNF-release from myeloid cells via ADAM17 can be induced the bacterial cell wall component lipopolysaccharide (LPS) [43,44] through toll-like receptor (TLR) 4 engagement [45]. During chronic liver disease, increased translocation of gut microbial products is enhanced and TLRs on KCs activated via pathogen-associated molecular patterns (PAMPs). Identification of TLR4 single-nucleotide polymorphisms [46] and use of TLR4-deficient mice [47] have demonstrated the importance of TLR4 for the development of fibrotic disease. While experimental proof is still lacking, it is likely that myeloid ADAM17 is essential for the establishment of hepatic inflammation and the development of liver fibrosis.

HSCs release EGFR ligands to promote hepatocyte proliferation [3]. Autocrine EGFR activation on KCs was recently identified to be critical for the induction of interleukin-6 (IL-6) release in a murine HCC model [48]. It is therefore conceivable that EGFR ligand release from HSCs induces IL-6 release from myeloid cells. On target cells, IL-6 binds to the membrane-bound IL-6 receptor (IL-6R) α, before engaging a gp130 homodimer that transduces the signal into the cell. This process is called “IL-6 classic signalling”. The proteolytic release of the soluble IL-6 receptor (sIL-6R) from KCs and infiltrated myeloid cells which is most likely mediated by ADAM17, in conjunction with secreted IL-6 creates a strong regenerative signal for hepatocytes [49,50]. On target cells, the complex of IL-6/sIL-6R can bind and therefore activate gp130, even on cells that do not express the membrane-bound IL-6R. This process is termed “IL-6 trans-signalling” [51]. Persistent activation of IL-6 trans-signalling on damaged hepatocytes can ultimately lead to the development of hepatocellular carcinoma (HCC) [52]. On the other hand, it has been shown that ectodomain shedding of c-Met is mediated by ADAM10 [53]. While experimental evidence is still lacking, it is conceivable that the overwhelming hepatocyte growth factor (HGF)-mediated mitogenic response of hepatocytes is dampened by ADAM10. This is supported by the fact that genetic c-Met-deficiency accelerates fibrosis in a CCl4-model [54]. We previously showed that ADAM10 is essential for hepatocyte homeostasis. Consequently, loss of hepatic ADAM10 resulted in spontaneous development of liver fibrosis [55]. This may, at least in part, be due to an accumulation of liver progenitor cells [55] that have previously been linked to activation of HSCs [56,57]. Furthermore, release of TNFα, presumably via ADAM17, sustains inflammation and promotes hepatocyte cell death [58].

Through the secretion of chemokine (C-C motif) ligand (CCL)2, activated KCs promote the CCR2-dependent recruitment of a pro-fibrotic CD11b^+^F4/80^+^Ly6C^hi^ macrophage population. ADAM protease-mediated release of soluble IL-6 receptor (sIL-6R) and induction of IL-6 trans-signalling furthermore induces migration of KCs/myeloid cells to the site of liver damage [52]. Secretion of pro-inflammatory and pro-fibrotic mediators by infiltrated myeloid cells promotes activation of HSCs [2]. Experimental depletion of CD11b^+^ cells in murine models of liver fibrosis has demonstrated that monocyte recruitment is essential for liver fibrosis [59,60].

Activation of HSCs and trans-differentiation into myofibroblasts is a key event during fibrosis of the liver. It is now a well-accepted concept that HSC activation and fate can be divided into (1) initiation, (2) perpetuation, and (3) clearance. During initiation, HSCs are rendered responsive to many extracellular signals e.g., through rapid up-regulation of receptor molecules. Perpetuation refers to events that sustain and amplify HSC activation. During clearance activated HSCs are removed either through induction of HSC apoptosis or the reversal into an inactive, quiescent state [3].

Transforming growth factor (TGF) β secreted from pro-fibrotic macrophages is the strongest inducer of HSC activation. Engulfment of epithelial cell apoptotic bodies by HSCs or DAMPs also contributes to HSC activation [61].

Rapidly after initial activation, HSCs up-regulate platelet-derived growth factor (PDGF) and its corresponding PDGF receptor β. PDGF is a strong HSC mitogen and promotes HSC proliferation [3].

### 3.2. Expression and Activation of ADAM Proteases on HSCs

Different studies have revealed that expression of several ADAM proteases, notably ADAM8, 9, 10, 12, 17, and 28, increased with HSC activation and that expression of these ADAMs was also detectable in fibrotic liver disease [62,63,64,65].

ADAM12 but not ADAM9 expression was demonstrated to be regulated by TGFβ in a phosphatidylinositol 3-kinase (PI3K) and mitogen activated protein kinase (MAPK)/extracellular signal regulated kinase (ERK) kinase (MEK)-dependent manner [62]. Data on TGFβ-dependency of the other HSC-associated ADAM family members is still lacking. Subcellular translocation of ADAM12 to the plasma membrane of HSCs was mediated downstream of integrin β1 binding to collagen I and subsequent intracellular activation of protein kinase C (PKC) ε (Figure 2a) [66,67]. Translocation of ADAM12 required its cytoplasmic tail that mediated binding to the scaffolding protein receptor of activated protein kinase C (ROCK) 1 [67].

Activation of ADAM17 on activated HSCs can occur via multiple pathways (Figure 2a). Advanced glycation end products (AGEs) that accumulate in diabetes patients, induce ADAM17 expression through NADPH oxidase 2 (NOX2). This is paralleled by a down-regulation of the natural ADAM17 inhibitor tissue inhibitor of metalloproteinase (TIMP) 3 [68]. Angiotensin II receptor AT1 is a GPCR associated with αq/11, and angiotensin II was demonstrated to elevate ADAM17 proteolytic activity on an HSC cell line [69]. Other GPCRs that are linked to an αq/11 subunit such as C-C chemokine receptor (CCR) 2, 5-hyroxytryptamine receptor (5-HT) 2 A, and proteinase activated receptor (PAR) 2 are expressed on HSCs [2], and engagement of these receptors might lead to downstream activation of PKC and ADAM17 and potentially other ADAM proteases. Also, PDGFR signalling via activation of the cytoplasmic kinases c-Src and ERK1/2 was demonstrated to induce ADAM17 activity [70]. However, it remains elusive as to whether PDGF signalling induces ADAM17 activity on HSCs as well.

### 3.3. Pro-Fibrotic ADAM Signals

Several ADAM proteases were demonstrated to support pro-fibrotic signalling pathways (Figure 2b). A common denominator of ADAMs 9, 10, 12, and 17 is the release of ligands for the epidermal growth factor (EGF) receptor. Increased expression of EGF was previously found in a rat model of cirrhosis [71] and inhibition of EGFR by the small molecular compound erlotinib decreased the number of activated HSCs and alleviated disease burden in three different model of progressive liver cirrhosis [72]. However, more in-depth analysis of different EGFR ligands revealed pro- and anti-fibrotic activities of EGFR signalling, depending on the ligand. While heparin-binding EGF-like growth factor (HB-EGF) was identified to suppress fibrosis (see also below) [73,74], amphiregulin (AR) was shown to promote liver fibrosis via direct induction of HSC activation and proliferation [65,75].

This selective and opposing effect of EGFR ligands is most likely due to their difference in receptor usage and a difference in ligand-dependent EGFR conformers. The EGFR family consists of EGFR (also termed ErbB1), ErbB2, ErbB3 (which lacks kinase activity), and ErbB4. While AR selectively binds to EGFR, HB-EGF has been shown to engage both EGFR and ErbB4 [76]. In contrast to EGFR, where ten tyrosine phosphorylation sites were identified, ErbB4 can be phosphorylated on 19 different tyrosine residues. As a consequence, ErbB4 downstream signalling differs from EGFR signalling [77]. Furthermore, downstream signalling of the EGFR itself is selective for the bound ligand. While HB-EGF induces a sharp peak of signalling terminated by receptor internalisation and down-regulation, AR has been shown to promote receptor recycling and sustained signalling, resulting in cellular proliferation [78]. This might be linked to the fact that AR does not induce EGFR Tyr1045 phosphorylation which is a binding site for the E3 ubiquitin ligase c-Cbl [77]. How can this ligand-dependent signalling selectivity be explained? Upon binding of EGF to the to the extracellular domain of the EGFR and consequent receptor dimerization, the N-terminal parts of the two juxtamembrane (JM) domains form an anti-parallel helical dimer [79]. While upon EGF and HB-EGF binding the JM dimer is formed by hydrophobic interactions, AR and other EGFR ligands induce JM dimer formation via polar interactions [78]. As a consequence, orientation of the two kinase domains to each other in an asymmetric dimer is slightly different which has been discussed to result in efficient phosphorylation of differential sets of tyrosine residues and subsequent differential activation of downstream signalling modules (Figure 2c) [77,78].

AR is a strong mitogen for hepatocytes [80,81] and HSCs [65,75]. Proteolytic release of AR and EGFR trans-activation was mediated by ADAM17 in an HSC cell line [69], while up-regulation of AR expression in fibrotic liver disease was paralleled by induction of ADAM17 expression [65], suggesting that ADAM17 could be the major protease for AR in the liver. Release of TNFα from myeloid cells, presumably by ADAM17, promoted survival of HSCs and contributed to the maintenance of fibrotic processes [61]. Furthermore, it was shown that ADAM17-induced IL-6 trans-signalling on macrophages up-regulated expression of the pro-fibrotic cytokine osteopontin [82]. Consequently, restoration of TIMP3 expression in mouse models of non-alcoholic steatohepatitis (NASH) reduced ADAM17 activity and lowered fibrotic activity in the liver, suggesting that ADAM17 is essential for NASH pathology [68].

ADAM12 seems to play an essential role during liver fibrosis and progression to cirrhosis. Its implication in fibrotic disease of other organs was demonstrated [83,84,85] and therefore ADAM12 seems to be part of a fibrotic core pathway, i.e., that is common to all fibrotic diseases. It enhanced HSC activation by two means. Independent of its cytoplasmic tail and proteolytic activity, ADAM12 bound to TGFβ receptor II (TβRII) and enhanced TβRII endocytosis and activation of Smad signalling from within early endosomes [86]. ADAM12 might therefore be an amplifier of TGFβ signalling. Furthermore, through its association with integrin-linked kinase (ILK), ADAM12 mediated β1 integrin signalling to Akt and cell sion-dependent survival. Accordingly, depletion of ADAM12 led to cytoskeletal rearrangements and impaired HSC adhesion [87].

Additionally, there is evidence that ADAMs are able to degrade extracellular matrix (ECM) components and therefore promote ECM rearrangement during wound healing and fibrosis in the liver. During liver fibrosis, type IV collagen and different proteoglycans are replaced with fibril-forming collagen type I and III [2]. ADAM9 and ADAM10 were shown to proteolytically release collagen XVII from the plasma membrane [88,89], while type IV collagen can be degraded by ADAM10 [90] and secreted ADAM12 [91]. None of the ADAMs have been shown to degrade type I collagen that accumulates during liver fibrosis. Fibronectin can be degraded by ADAM9 [92], soluble ADAM12 [91], and ADAM13 [93], and laminin has been shown to be processed by soluble ADAM9 [94].

### 3.4. Anti-Fibrotic ADAM Signals

Beside its pro-fibrotic role, ADAM proteases might also be implicated in the termination of pro-fibrotic pathways.

HB-EGF released from KCs/macrophages is able to prevent liver fibrosis by two means. On one hand, HB-EGF might enhance hepatocyte fitness through KC/macrophage autocrine EGFR activation and release of IL-6 [39], and enhanced DNA damage repair within hepatocytes [86]. On the other hand, HB-EGF was demonstrated to directly suppress TGFβ-induced expression of collagen I in HSCs [64,65]. ADAM proteases might therefore fine-tune disease progression through differential release of EGFR ligands. While detailed data in the liver are lacking, ADAM9, 12, and 17 were identified as HB-EGF sheddases [87,88,89] and may therefore also be responsible for HB-EGF release from KCs and activated HSCs.

Downstream of ERK-activation, ADAM17 was identified to promote proteolytic processing of TGFβ receptor, therefore blunting TGFβ signalling [95]. It is therefore possible that ADAM17 on HSCs not only promotes pro-fibrotic pathways but also contributes to prevent overshooting HSC activation. However, direct experimental proof is still lacking.

CD11b+Ly6Clo restorative macrophages that arise from CD11b+Ly6Chi inflammatory liver-infiltrating macrophages express the CX3C chemokine receptor 1 (CX3CR1). It was demonstrated that binding of its ligand CX3CL1 (also known as fractalkine) inhibited pro-inflammatory activities and promoted fibrosis resolution through secretion of matrix metalloproteinases (MMPs) and induction of HSC apoptosis through TNF-related apoptosis-inducing ligand (TRAIL) [2]. Both ADAM10 and ADAM17 were identified in in vitro studies to release CX3CL1 from HSCs and promote monocyte migration [64]. However, this study did not clarify whether ADAM-mediated release of CX3CL1 was pro- or anti-fibrotic.

## 4. Therapeutic Perspectives

Current therapeutic strategies that directly target hepatic stellate cell biology have entered phase II and III trials and include the targets peroxisome proliferator-activated receptor (PPAR) α, PPARγ, PPARδ, farnesoid-X receptor (FXR), and chemokine receptors such as CCR2 or CCR5 [2,96]. Given the fact that both CCR2 and CCR5 are linked to an αq/11 subunit, inhibition of these receptors might also lower ADAM protease activity. Interestingly, the ADAM10 promoter contains a number of PPAR response elements (PPREs) [97]. All PPARs heterodimerize with the retinoid-X-receptor (RXR), and it has been demonstrated that both retinoic acid and PPARα-agonists increase ADAM10 expression [97,98]. Alterations of ADAM-mediated signalling pathways by HSC-targeting therapies should therefore be considered in future studies.

Furthermore, there is evidence that selective interference with ADAM proteases might represent a novel therapeutic approach to target liver fibrosis. Serum levels of soluble Axl were elevated in hepatitis C virus patients and patients with alcoholic liver disease [99]. Ectodomain shedding of the transmembrane receptor tyrosine kinase Axl from systemic lupus erythematosus (SLE) leukocytes was mediated by ADAM10 and 17 in systemic lupus erythematosus [100] and it is conceivable that sAxl levels during acute and chronic liver disease correlate with ADAM10/17 activity in the liver. Soluble ectodomains in the serum therefore might be considered as surrogate markers for hepatic ADAM protease activity. However, a more systematic and in depth analysis is needed to map liver fibrosis-associated ADAM substrate release to disease progression. Furthermore, diagnosis of hepatic fibrosis by non-invasive imaging techniques might be improved through the use of labelled ADAM substrate molecules or ADAM-selective activity probes. Such approaches have been used for successful monitoring of, e.g., caspase activity by magnetic resonance imaging (MRI) [101,102].

As indicated above, the different ADAM family members seem to play differential roles during hepatic fibrosis. ADAM-selective inhibition might therefore skew development of disease towards fibrosis resolution, such as e.g., selective release of EGFR ligands. Given its prominent role in hepatic and other tissue fibrosis, selective inhibition of ADAM12 may be a promising approach.

Selective inhibition of ADAM proteases is a challenging task. Previous approaches to inhibit ADAM17 for the therapy of chronic inflammatory diseases have failed due to severe musculoskeletal and hepatic side effects of the compounds due to incomplete specificity and a large substrate spectrum [103]. The pro-domains of ADAM proteases share only a low level of similarity and the use of recombinant forms of ADAM pro-domains has been considered as an alternative approach to specifically inhibit ADAM proteases. Recombinantly produced pro-domains of ADAM10, 12, and 17 were demonstrated to have inhibitory selectivity for their cognate metalloproteases in vitro [104,105,106,107]. The recombinant pro-domain of ADAM17 was further used for specific ADAM17 inhibition, including a model of kidney fibrosis, where it was demonstrated to inhibit ADAM17-mediated release of AR [107,108,109,110,111]. This concept might be expanded to other ADAM family members and its use in murine models of hepatic fibrosis and cirrhosis will demonstrate its therapeutic potential in the future.

Recent crystallographic data on ADAM10 has revealed that the disintegrin domain and the cysteine-rich domain of ADAM10 form an auto-inhibitory module folding back to the catalytic site in order form a closed conformation. Furthermore, the cysteine-rich domain of all human ADAMs contain a hyper-variable region with low sequence homology to other ADAMs. Stabilisation of the closed conformation of ADAMs using specific antibodies might therefore represent an attractive mean to inhibit ADAM proteases. It is possible that targeting the hyper-variable region might increase antibody specificity in this context.

On the other hand, antibodies or small peptides that interfere with the interaction of cysteine-rich domain and catalytic domain represent an attractive way to enhance ADAM protease activity.

Targeting the interaction of ADAM proteases with its regulatory proteins such as iRhoms or Tspans represents another way to specifically target ADAM proteases. As an example, inhibition of iRhom2/ADAM17 interaction would be a specific means of preventing ADAM17-mediated release of inflammatory signals from myeloid cells without affecting essential ADAM17 activity in other tissues.

In summary, the family of ADAM proteases remains a rich but largely under-explored sea of therapeutic opportunities for the treatment of liver fibrosis.

## Figures and Tables

**Figure 1 cells-08-01226-f001:**
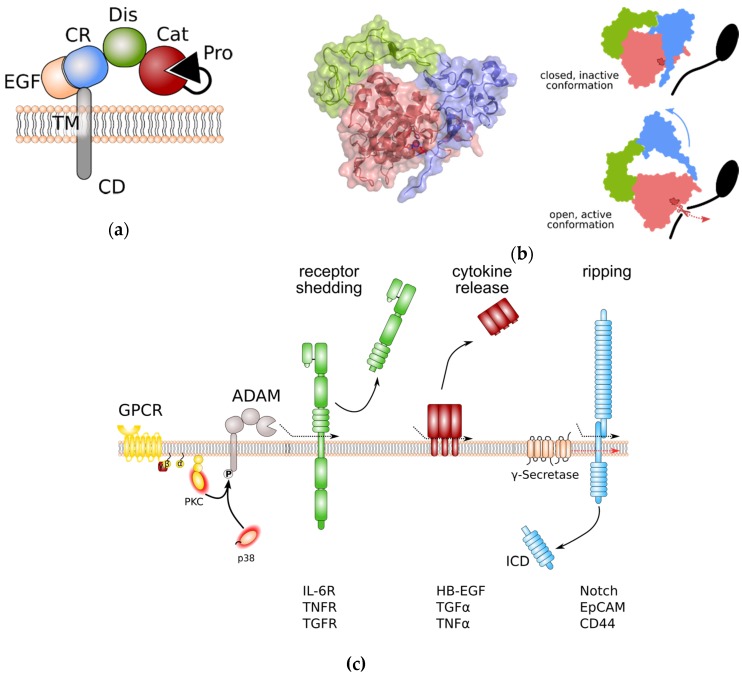
General structure of a disintegrin and metalloprotease (ADAM) proteases and overview of ADAM protease-induced pathways. (**a**) Domain structure of ADAM proteases. (**b**) Left panel: Crystal structure of the ADAM10 closed conformation, including the catalytic, disintegrin, and cysteine-rich domain. The Zn^2+^-binding motif is highlighted in dark red. Colouring of domains is as in (**a**). Note that the cysteine-rich domain prevents access to the catalytic site. Right panel: Anticipated movement of ADAM10 domains during activation. Colour coding as in (**a**). The Zn^2+^-binding motif of the catalytic site is indicated in dark red. (**c**) General signalling pathways regulated by ADAM proteases. Protein phosphorylation can regulate ADAM protease activity. Limited proteolysis by active ADAM proteases induces removal of receptor ectodomains, release of membrane-bound cytokines and growth factors, and initiates signal transduction by regulated intramembrane proteolysis (RIP). Removal receptor ectodomain can either blunt receptor signalling or induce so-called “trans-signalling” on neighbouring cells. EpCAM: epithelial cell adhesion molecule; EGF: epidermal growth factor; GPCR: G-protein coupled receptor; HB-EGF: heparin-binding EGF-like growth factor; ICD: intracellular domain; IL-6R: interleukin-6 receptor; PKC: protein kinase C; TNFα: tumour necrosis factor α; TNFR: TNFα receptor; TGFα: transforming growth factor α; TGFR: TGFβ receptor.

**Figure 2 cells-08-01226-f002:**
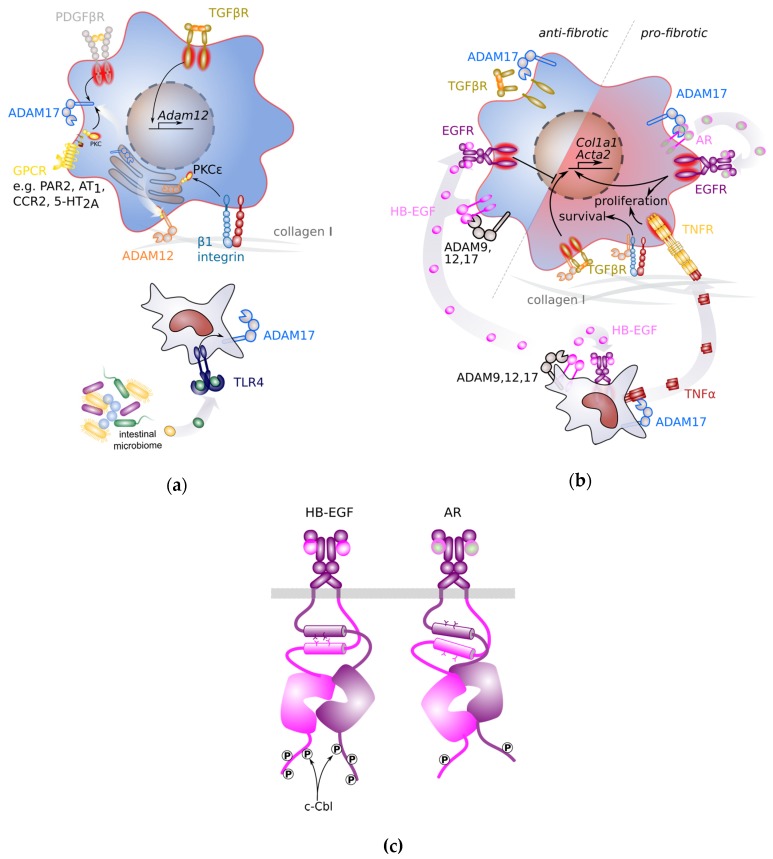
Major a disintegrin and metalloprotease (ADAM)-associated fibrotic pathways. (**a**) Molecular pathways leading to expression and activation of ADAM proteases in hepatic stellate cells (HSCs) or myeloid cells. (**b**) Pro- and anti-fibrotic pathways regulated by ADAM proteases. (**c**) Heparin-binding EGF-like growth factor (HB-EGF) and amphiregulin (AR) binding to epidermal growth factor receptor (EGFR) induces different EGFR conformers, resulting in the activation of different downstream signalling pathways such as recruitment of E3 ubiquitin-protein ligase Casitas B-lineage lymphoma (c-Cbl) in HB-EGF-stimulated but not AR-stimulated receptors. Schematic adapted from [78]. Differential down-stream signalling of EGFR in HSCs might account for the difference in fibrotic activity of AR and HB-EGF. ADAM: a disintegrin and metalloprotease; AR: amphiregulin; AT1: angiotensin II receptor 1; β1 integrin; CCR2:; EGFR: epidermal growth factor receptor; GPCR: G-protein coupled receptor; 5-HT2A: 5-hydroxytryptamin receptor 2A; HB-EGF: heparin-binding epidermal growth factor; PAR2: proteinase activated receptor 2; PDGFβR: platelet-derived growth factor β receptor; PKC: protein kinase C; TGFβR: transforming growth factor β receptor; TLR4: toll-like receptor 4; TNFα: tumour necrosis factor α; TNFR: TNF receptor.

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
