# Peer review of "Regulation of Fibrotic Processes in the Liver by ADAM Proteases"

_cells, 2019, doi:10.3390/cells8101226_

Round 1

Reviewer 1 Report

The manuscript ”Regulation of fibrotic processes in the liver by ADAM proteases” by Schmidt-Arras and Rose-John provides an excellent overview of the role of ADAM proteases in fibrotic liver disease. Given the strong evidence of an important role of ADAMs in liver fibrosis and the lack of any recent reviews on this topic, this is a timely and valuable contribution to the field of research.

The authors are recommended to address a few minor comments:

The abbreviation NKT (line 25) should be spelled out.

On line 55-60, the authors describe the furin-mediated pro-protein processing of ADAM proteases, mentioning the fact that ADAM17 furin-resistant variants retain the capacity to shed TNFa. Given that 2 consecutive ADAM17 pro-protein cleavage sites have been reported, it is unclear whether both sites were disrupted or only one, in which case the remaining site could potentially take over. Please specify.

The description of ADAM regulation on p. 4 is lacking important information on endocytic regulation of ADAMs. In this respect, the possibility of targeting ADAM subcellular localization could even be discussed in the section on Therapeutic perspectives.

The last paragraph of Section 2 (lines 147-152) seems a little misplaced. It is suggested to include this information in the beginning of Section 3.

The anti-fibrotic effect of HB-EGF, described on line 275-279 could be moved to Section 4 anti-fibrotic ADAM signals

Author Response

Reviewer 1:

The manuscript ”Regulation of fibrotic processes in the liver by ADAM proteases” by Schmidt-Arras and Rose-John provides an excellent overview of the role of ADAM proteases in fibrotic liver disease. Given the strong evidence of an important role of ADAMs in liver fibrosis and the lack of any recent reviews on this topic, this is a timely and valuable contribution to the field of research.

The authors are recommended to address a few minor comments:

The abbreviation NKT (line 25) should be spelled out.

We have spelled out NKT in the new version. The sentence reads now like:

HSCs are discussed to be professional antigen presenting cells, in particular presenting lipid antigens to natural killer T-(NKT) cells [1, 2].“

On line 55-60, the authors describe the furin-mediated pro-protein processing of ADAM proteases, mentioning the fact that ADAM17 furin-resistant variants retain the capacity to shed TNFa. Given that 2 consecutive ADAM17 pro-protein cleavage sites have been reported, it is unclear whether both sites were disrupted or only one, in which case the remaining site could potentially take over. Please specify.

We thank the reviewer for pointing out this unclarity in our text. We added additional information and an additional reference. The paragraph reads now like:

Interestingly, for ADAM9, 10 and 17 an additional pro-protein convertase cleavage site (upstream site) N-terminal to the previously identified Furin cleavage site (boundary site) was discovered [11]. While cleavage of the C-terminal boundary site by Furin was sufficient for pro-domain removal but dispensable for protease activity, as observed in our experiments, cleavage at the N-terminal upstream site was necessary for the dissociation of the pro-domain and full activation of ADAM proteases [11].

The description of ADAM regulation on p. 4 is lacking important information on endocytic regulation of ADAMs. In this respect, the possibility of targeting ADAM subcellular localization could even be discussed in the section on Therapeutic perspectives.

We thank the reviewer for spotting this lack of information in our text. We added an additional paragraph discussing internalisation of ADAM proteases which reads like:

The C-terminus of ADAM proteases is furthermore important for their negative regulation via endocytosis. Binding of the cytosolic adaptor proteins Grb2 to ADAM12 and AP2 to ADAM10 were demonstrated to mediate internalisation via the clathrin-dependent pathway [34, 35]. Interestingly, overexpression of dominant negative dynamin K44A prevented internalisation but enhanced proteolytic processing of surface-bound ADAM10 [36]. ADAM17 was also shown to be internalised upon phorbol-12-myristate-13-acetate (PMA)-stimulation [37]. Complex formation of ADAM17 with iRhoms and the FERM domain-containingprotein (FRMD) 8 at the cell surface was identified to prevent ADAM17 internalisation and lysosomal degradation [38, 39]. Once internalised, interaction of ADAM17 with the sorting protein phosphofurin acidic cluster sorting protein (PACS) 2 at early endosomes promotes recycling of ADAM17 back to the cell surface [40].“

The last paragraph of Section 2 (lines 147-152) seems a little misplaced. It is suggested to include this information in the beginning of Section 3.

We agree with the reviewer and have moved this paragraph as suggested.

The anti-fibrotic effect of HB-EGF, described on line 275-279 could be moved to Section 4 anti-fibrotic ADAM signals

We agree with the reviewer and moved this paragraph accordingly.

Reviewer 2 Report

This is a well written review article.

I have no major comments.

I found that one piece of work that might be worth of mentioning is the effect of   bile acids and their receptors on  HSC and ADAM as well as PPRs, PXR and so on.

Also CCL2 and other chemokines could be of interest.

The above on the light  that these  drug targets are close to  reach approval for   clinical application.

Author Response

Reviewer 2:

This is a well written review article.

I have no major comments.

We are grateful to the reviewer for this assessment.

I found that one piece of work that might be worth of mentioning is the effect of   bile acids and their receptors on  HSC and ADAM as well as PPRs, PXR and so on.

Also CCL2 and other chemokines could be of interest.

The above on the light  that these  drug targets are close to  reach approval for   clinical application.

We thank the reviewer for this important suggestion. We now included a paragraph on regulation of ADAM proteases by bile acids in section 2, which reads like:

There is some evidence that in the liver, ADAM protease activity is regulated by bile acids. Ursodeoxycholic acid was shown to inhibit PMA-induced ADAM17 activity on both, the HepG2 hepatocellular cell line, as well as the hepatic stellate cell line LX-2 [41]. In contrast, deoxycholic acid (DCA) enhanced ADAM17 catalytic activity and release of amphiregulin in colorectal and pancreatic cancer cell lines. This effect was mediated via binding of DCA to the G protein-coupled bile acid receptor 1, also termed TGR5, and subsequent c-Src activation [42].“

We furthermore inserted a paragraph in section 4 which links current therapeutical approaches to ADAM proteases and which reads like:

Current therapeutical strategies that directly target hepatic stellate biology entered phase II and III trials and include the targets peroxisome proliferator-activated receptor (PPAR) α, PPARγ, PPARδ, farnesoid-X receptor (FXR) and chemokine receptors such as CCR2 or CCR5 [2, 96]. Given the fact, that both, CCR2 and CCR5 are linked to an αq/11 subunit, inhibition of these receptors might also lower ADAM protease activity. Interestingly, the ADAM10 promoter contains a number of PPAR response elements (PPRE) [97]. All PPARs heterodimerize with the retinoid-X-receptor (RXR) and it has been demonstrated that both, retinoic acid, as well as PPARα-agonists increase ADAM10 expression [97, 98]. Alterations of ADAM-mediated signalling pathways by HSC-targeting therapies should therefore be considered in future studies.

Furthermore, there is evidence that selective interference with ADAM proteases might represent a novel therapeutic approach to target liver fibrosis.“